# Solidification of Self-Emulsifying Drug Delivery Systems as a Novel Approach to the Management of Uncomplicated Malaria

**DOI:** 10.3390/ph15020120

**Published:** 2022-01-20

**Authors:** Eun Bin Seo, Lissinda H. du Plessis, Joe M. Viljoen

**Affiliations:** Centre of Excellence for Pharmaceutical Sciences (Pharmacen^TM^), Building G16, North-West University, 11 Hoffman Street, Potchefstroom 2520, South Africa; sunnygoed@yahoo.com (E.B.S.); Lissinda.DuPlessis@nwu.ac.za (L.H.d.P.)

**Keywords:** solidification techniques, self-emulsifying drug delivery systems, malaria, artemisinin-based combination therapy, artemether, lumefantrine, fixed-dose combination, lipid-based formulations, lipophilic drug delivery

## Abstract

Malaria affects millions of people annually, especially in third-world countries. The mainstay of treatment is oral anti-malarial drugs and vaccination. An increase in resistant strains of malaria parasites to most of the current anti-malarial drugs adds to the global burden. Moreover, existing and new anti-malarial drugs are hampered by significantly poor aqueous solubility and low permeability, resulting in low oral bioavailability and patient noncompliance. Lipid formulations are commonly used to increase solubility and efficacy and decrease toxicity. The present review discusses the findings from studies focusing on specialised oral lipophilic drug delivery systems, including self-emulsifying drug delivery systems (SEDDSs). SEDDSs facilitate the spontaneous formation of liquid emulsions that effectively solubilise the incorporated drugs into the gastrointestinal tract and thereby improve the absorption of poorly-soluble anti-malaria drugs. However, traditional SEDDSs are normally in liquid dosage forms, which are delivered orally to the site of absorption, and are hampered by poor stability. This paper discusses novel solidification techniques that can easily and economically be up-scaled due to already existing industrial equipment that could be utilised. This method could, furthermore, improve product stability and patient compliance. The possible impact that solid oral SEDDSs can play in the fight against malaria is highlighted.

## 1. The Perpetual Battle with Malaria

Despite intensive efforts to control malaria, it remains a devastating global challenge. Malaria is endemic to 97 countries and a prevalent vector-transmitted disease worldwide. According to the World Health Organization’s (WHO) malaria report, a projected 229 million cases of malaria occurred internationally in 2019, with the WHO African Region accounting for 94% of all malaria deaths. Globally, the mortality rate ranges from 0.3–2.2%, whereas regions with tropical climates depict mortality statistics of 11–30% [1,2].

Malaria is caused by *Plasmodium* parasites that are commonly transmitted through bites of the female *Anopheles* mosquito. Upon arrival in human blood, the parasites occur as sporozoites, which enter hepatocytes within thirty minutes after release into the systemic circulation. Following this, an asexual developmental phase arises in the erythrocytes where all *Plasmodium* species induce erythrocyte rupture [1]. Normally, malaria is the suspected cause of disease primarily based on a patient presenting with a history of fever or a temperature exceeding 37.5 °C that has visited or resides in a malaria-endemic area [3].

The *Plasmodium falciparum* and *Plasmodium vivax* species are responsible for the majority of malaria-related deaths. Transmission of this disease is severe in areas where the mosquitos have strong human-biting habits and longer lifespans [4]. Moreover, a growing number of different possibilities of infection exist. A particular malaria infection process, defined as “Odyssean”,” mini-bus” or “baggage” malaria, has recently been documented. This process allows the transmission of, especially, the *P. falciparum* parasite to non-endemic areas as a result of its ability to survive inside passengers’ baggage under favourable climatic conditions [5,6]. Additionally, “airport malaria” may result from the infective bite of a tropical anopheline mosquito that has travelled via plane (sometimes even internationally). After the mosquito exits the plane, the likelihood of survival is high before a blood meal is needed. The temperate and humid environments of unlikely regions during the summer months can be favourable for survival and reproduction. For these reasons, myriads of travellers from developed countries are becoming increasingly more likely to contract or spread malaria each year. In addition, both of these routes of transmission have a correlation with the frequently delayed diagnosis of affected patients, rendering this disease even more dangerous [7,8].

Apart from the extensive symptoms such as fever, headache, vomiting, diarrhoea, chills and muscle pain, malaria also has indirect effects. These can emerge before birth through malarial pregnancies, leading to miscarriages, neonatal and infant mortality, deformities and congenital infections. It can also increase the risk of other infections. Furthermore, chronic malaria is strongly associated with hyperactive malarial splenomegaly, chronic renal damage, nephrotic syndrome, Burkitt’s lymphoma and anaemia—these disorders directly affect work productivity and quality of life [9].

The deteriorating political and economic stability in many developing countries, particularly in Africa, is compromising the medical treatment of malaria. This relationship between malaria and poverty is a malicious cycle. The WHO classifies malaria as one of the “big three” diseases of poverty together with HIV/AIDS and tuberculosis [10]. Poverty-related diseases are the leading cause of death in children and adolescents and primarily affects low- and medium-income countries where they cause a significant burden on the population’s health and have a considerable negative impact on the already overwhelmed economic development. Diseases such as malaria directly affect the most vulnerable populations—those with limited access to adequate health care services and acceptable nutritional diets—resulting in a notably persistent threat to global health [11]. Hence, it is clear that malaria does not only impact the affected individuals but has a negative effect on all those around them, including the general society. It is also clear that malaria is not only an African or tropical region problem, but is becoming a global threat. This work consequently demonstrates that numerous issues can be resolved, utilising well-established oral drug delivery systems that are applied in a novel manner; i.e., the successful utilization of solidified self-emulsifying drug delivery systems (SEDDSs), which could thus potentially significantly improve current malarial treatment regimens, ensuring increased patient compliance in especially poverty-stricken regions, without increasing the economic burden of patients.

## 2. Treatment Regimens Currently Suggested for Uncomplicated Malaria

The main clinical objectives of the WHO [3] for the management of uncomplicated malaria are to treat the infection as swiftly as possible while avoiding its advancement to a more severe degree of disease. For this reason, it is imperative to start effective management as soon as possible by means of rapid diagnosis and the initiation of the correct therapy. The Swiss Tropical and Public Health Institute has published malaria treatment recommendations in 2020, which suggest various treatment regimens for the effective handling of uncomplicated malaria in general [12]. Interestingly, monotherapy with drugs such as amodiaquine, chloroquine and mefloquine is still recommended as the first-line treatment of *P. cynomolgi*, *P. malariae*, *P. ovale* and *P. vivax* (Table 1).

Alternatively, artemisinin-based combination therapies (ACTs) are often deemed as the most successful treatment approach to timely diagnosed, uncomplicated *P. falciparum* malaria as they have been shown to significantly reduce the mortality rate of this type of malaria over the past few years [4]. This combination includes a fast-acting artemisinin derivative together with a longer-acting and notably slower eliminated partner constituent. The artemisinin component rapidly clears parasites from the blood, reducing parasitic numbers by a factor of approximately 10,000 in each 48 h asexual cycle [13]. Additionally, it is active against the sexual stages of the gametocytes that mediate onward transmission to mosquitos [14]. The role of the longer-acting companion drug, on the other hand, is to clear the residual organisms and afford protection against the progression of resistance to the artemisinin derivative. Partnering drugs with increased elimination half-lives may also provide a further post-treatment prophylaxis period [3].

The five ACTs generally recommended treating children and adults (except antenatal women in the first trimester) affected by uncomplicated *P. falciparum* malaria according to the Guidelines for the Treatment of Malaria [3] are summarised in Table 2, where the existing available commercial products are also listed [15].

### 2.1. Why Fixed-Dose Combination Therapy

The concept of fixed-dose combination (FDC) therapy consists of two or more active ingredients within a single dosage form [16]. It is based on the synergistic or additive potential of two or more drugs to improve therapeutic efficacy, as well as to delay the development of resistance to the individual components of the combination. Achieving effective antimalarial drug concentrations for an adequate exposure time and circumventing drug resistance is critical in order to ensure high cure rates. For this reason, patient adherence is vital. It has been found that numerous patient groups, especially younger children, are not optimally treated with the recommended individual dosage regimens, thus compromising efficacy and fuelling resistance. FDC therapy offers an alternative means to evade malaria and other disease recrudescence in patients by simplifying the dosing regime and often lowering the dosages of the individual agents due to the synergistic potential offered [16,17].

For communicable diseases, such as malaria, adherence to dosing regimens may notably influence the attainment of positive healthcare outcomes [18]. As an example, FDC formulations for diabetes improved patient compliance, enriched consumer satisfaction and lowered medical expenditures by 10–13% [19]. Another study indicated that FDC therapy decreased the risk of non-adherence in patients with HIV, hypertension, tuberculosis and diabetes by approximately 25% [20]. However, one needs to understand the complex relationship between the drug dose and pharmacokinetics, as well as the consequential pharmacodynamics (therapeutic efficacy) and drug safety [3,16]. Additionally, issues regarding cost and access have also been associated with FDC treatments. Nonetheless, there exists a plausible likelihood for future improved availability and use of FDC therapies across lower- and middle-income countries [16].

Combination therapy for uncomplicated malaria was first suggested for sub-Saharan Africa in the year 2000 [21]. Nine years later, South Africa’s Minister of Health released Guidelines for the Treatment of Malaria, where the artemether-lumefantrine (ART-LUM) combination (Coartem^®^), or alternatively, quinine and either doxycycline or clindamycin were recommended for the treatment of uncomplicated malaria [22,23]. ACTs have been endorsed by the WHO as the first-line treatment of uncomplicated *P. falciparum* malaria since 2001. Concurrently, more than 80 countries have applied these treatment regimens where most populations have used them for more than a decade already. Moreover, in 2007, in Côte d’Ivoire, the National Malaria Control Programme (NMCP) undertook to assure provision and distribution of ACTs in all malaria treatment centres throughout this state [24,25,26,27,28]. The reasoning behind the use of ACTs is that it is based on the simultaneous rapid elimination of asexual parasitemia, together with the alteration of the sexual stages of gametophytes, subsequently reducing malaria infectivity [29,30,31]. Gbotosho et al. [29] moreover suggested that ACT treatment could transfer immature gametocytes from the bone marrow to the peripheral bloodstream and promptly clear them from the human body.

Artemisinin and its derivatives work rapidly. However, their markedly short half-lives warrant mono-therapeutic courses for *P. vivax* and *P. falciparum* malaria of five and seven days, respectively. These are meaningfully long course durations that extensively complicate therapy with artemisinin (and/or its derivatives) in malaria-endemic regions, including Africa and Southeast Asia. To evade this problem, a second drug, usually possessing a longer half-life, is added to the dosage regime. This combination is able to shorten treatment to only 3 days and additionally ensures the elimination of any remaining parasites, while also preventing the development of drug resistance [27,32,33,34,35,36].

The ART-LUM combination (Coartem^®^) is currently the most widely used ACT commercially available treatment for especially uncomplicated *P. falciparum* malaria. Over 900 million patients, including more than 390 million paediatric patients, in malaria-endemic countries, have successfully used Coartem^®^ as first-line therapy [37,38]. It has been estimated that Coartem^®^ already constituted around 73% of global ACT procurements in 2013, rendering it one of the most extensively-used anti-infective therapies in the world at present [27,33,38].

### 2.2. Facing Artemisinin-Based Combination Therapy Shortcomings

In recent years, an increase in the number of multidrug-resistant forms of *P. falciparum* parasites has been observed during treatment, particularly when monotherapy was applied. For this reason, the WHO highly recommends the use of ACTs [3]. However, evidence does exist indicating that this type of therapy displays some shortcomings.

Firstly, due to the highly lipophilic properties and poor aqueous solubility of the said drugs, for example, the ART-LUM combination, absorption thereof is variable and food-dependent. Indeed, treatment failure has been associated with incomplete drug absorption [39,40]. In addition, a “fatty meal” is predominantly required to invoke the “food effect” [39,41,42,43,44]. This “food effect” may enhance the absorption of co-administered lipophilic drugs (Section 3.2) due to the stimulation of bile secretion from the gallbladder, which in turn establishes a solubilising milieu of acid and homogenised fat to stimulate dietary lipid absorption [45]. It has been reported that the oral bioavailability of LUM can be increased up to 16-fold when it is co-administered with lipids [41,46]. Stone et al. [47] concluded that maximum gallbladder contraction ensues when a quantity of 10–25 g of lipids have been consumed, rendering the required fat intake for enhanced drug absorbance significantly high. Conversely, Kossena and associates [43] have revealed that lipid quantities as low as 2 g are capable of stimulating sufficient amounts of bile salt to raise intraluminal lipid processing and drug solubilisation. Nonetheless, due to the typical gastrointestinal (GI) disturbances normally experienced with malaria, including abdominal pain, nausea, vomiting and loss of appetite, infected patients are reluctant to eat properly—especially fatty foods [41,48,49]. This leads to variable drug bioavailability, inadequate therapeutic drug-blood levels as well as patient noncompliance. In addition, due to the complexity of malaria in conjunction with poverty and vulnerable populations that have limited access to adequate health care services and acceptable nutritional diets, treatment failure poses a massive threat [11].

In 2017, a retrospective study from Sweden on non-immune travellers with uncomplicated *P. falciparum* malaria who received the standard 3-day ART-LUM treatment regimen depicted a notably high rate of late treatment failures (5.3%) [50]. Similarly, other retrospective and case studies in Belgium and the United Kingdom reported similar treatment failures with the same treatment regime [51,52]. Recrudescence was observed approximately 20–43 days post-completion of the treatment program in both studies; however, no evidence was found of underlying drug resistance to this ACT regimen. It was rather alleged that the low LUM plasma concentration detected in these patients was the culprit. Clinical efficacy is determined by LUM plasma concentrations on day seven of treatment as these concentrations are responsible for the eradication of residual parasites not eliminated by ART [53]. A prospective 2008 study on the same subject indicated a treatment failure rate of 5.3% in non-immune European travellers (unpublished subgroup analysis). Here, the authors argued that the treatment failure was due to under-dosing rather than incipient drug resistance [54]. It can, therefore, be debated that under-dosing may also occur due to varied and/or limited LUM absorption, which, in turn, most probably stems from an inadequate drug delivery system.

To emphasise, although a growing number of imported malaria treatment failures have been conveyed in the past five years, it could not be accredited to parasites carrying validated molecular signatures of ACT resistance, thus signifying that current treatment regimens remain mostly effective. However, more personalised management and revised regimens should be explored to warrant sufficient dosage [51].

## 3. Lipid Formulations: The Past and the Present

Nearly 70% of currently available drugs are reasoned to belong to Class II of the Biopharmaceutics Classification System (BCS). Hence, almost three-quarters of all drugs possess low solubility and high permeability—factors that impede oral drug absorption and subsequent bioavailability. Enhancing the ensuing drug solubility, and therefore, bioavailability is a multidimensional challenge [55]. Fortunately, numerous technologies are accessible to improve solubility and overcome bioavailability issues. Among these solubilisation techniques, lipid formulations have been extensively employed in accepted drug products since 1980 [56].

Lipid-based systems, often containing the esters of fatty acids, range from anything as simple as a drug in oil to much more complex formulations designed to spontaneously emulsify upon contact with aqueous media. These systems, which are listed below, are commercially viable to formulate safe and efficient pharmaceuticals for topical, oral, pulmonary or parenteral delivery [57]:Emulsions include microemulsions, nanoemulsions, self-emulsifying drug delivery systems (SEDDSs), self-micro-emulsifying drug delivery systems (SMEDDSs), self-nanoemulsifying drug delivery systems (SNEDDSs) and Pickering emulsions.Vesicular systems consist of liposomes, biosomes, pharmacosomes, phytosomes, transferosomes, ethosomes, archaeosomes, vesosomes, colloidosomes and herbosomes.Lipid particulate systems comprising lipospheres, solid-lipid particles, solid-lipid microparticles, solid-lipid nanoparticles, nanostructure lipid carriers and lipid drug conjugates [57,58,59].

Of these formulations, SEDDSs, utilising spontaneous emulsification, are well-known. Conventionally, traditional SEDDSs have been used in their original liquid form as oral drug delivery systems or to fill either soft or hard gelatin capsules. Lately, several solidification techniques have been explored to create more adequate delivery systems, including renovating liquid SEDDSs into dispersible powders, pellets and granules to enhance the stability of these uniquely simplified systems [60,61,62]. What is more, recent research has proven that SEDDSs may successfully enhance drug delivery via diverse pathways, such as the vaginal-, rectal-, ocular-, nasal and topical/transdermal routes of administration [63,64,65].

### 3.1. The Lipid Formulation Classification System (LFCS)

The Lipid Formulation Classification System (LFCS) was first introduced as a working model by Pouton in 2000, with an extra type of formulation being added during 2006. The purpose of this classification system is to efficiently interpret the in vivo behaviour in the identification of specific drugs. The LFCS categorises formulations into four groups (i.e., types I–IV) in accordance with their composition and the potential influence of dilution and digestion on their ability to prevent drug precipitation [66,67,68,69].

Type I systems are comprised of modest oil solutions (100% pure oil) without any surfactant or co-surfactant. These merely contain mono-, di- and/or tri-glycerides. They are non-dispersing and require digestion. Type II systems, on the other hand, consist of an oil phase (60–80%) with lipophilic surfactant(s) added to increase the solubilisation of the incorporated drug(s) as well as to assist with emulsion stability upon dilution. These types of formulations are classically labelled as SEDDSs. SMEDDSs and SNEDDSs (often used interchangeably) are both characterised as type III systems where more hydrophilic surfactants and/or co-solvents are combined. Type III systems are further separated into two types where distinction is made between the hydrophilic and lipophilic properties of the formulations. Type IIIA systems comprise about 40–60% oil and are more hydrophobic in nature, whereas those belonging to type IIIB consist of only 20–50% oil and include more hydrophilic surfactants and/or co-surfactants. This sub-class depicts higher dispersion rates. However, the risk of untimely drug precipitation upon dispersion is higher due to its low lipid content [70]. Lastly, type IV systems (most hydrophilic group) contain only hydrophilic surfactants and co-solvents. These formulations form colloidal micellar dispersions upon dilution with an aqueous medium. Considering these characterised formulation types, the LFCS is deliberated to be an easy approach to differentiate between SEDDS, SMEDDS and SNEDDS [67,71,72,73,74,75].

### 3.2. Optimising Oral Lipophilic Drug Delivery through Natural Solubilisation

The inclusion of lipophilic or poorly aqueous soluble drugs (i.e., BCS class II or IV drugs), for instance, ACTs, into a pharmaceutical formulation can present substantial formulation complications as these drugs display reduced absorption from an aqueous environment such as the gut [76]. As stated earlier, co-administration of lipophilic drugs with a fat-rich meal can improve drug bioavailability. For this reason, there has been increasing interest in the advancement of lipid-based oral formulations to improve drug solubilisation in the GI tract (GIT) [77]. In order to appreciate the benefits and mechanisms of action of lipid-based formulations, it is imperative to understand the body’s physiological response to lipid consumption.

The release of the intestinal hormone cholecystokinin into the duodenum is stimulated by the presence of chyme, which contains lipids. Cholecystokinin relaxes the hepatopancreatic sphincter and encourages contraction of the gallbladder, which in turn releases bile-containing bile salts, phospholipids (mostly phosphatidylcholine) and cholesterol into the small intestine. At the same time, pancreatic lipase/co-lipase (i.e., pancreatic fluids) are also released into the intestines [45,78]. These endogenous GI fluids possess an intrinsic ability to solubilise the ingested lipids. Digestion of lipids can, however, also occur to a lesser extent in the stomach via acid-stable lipases—the source of initial lipid emulsification. The process continues in the small intestine by means of pancreatic lipases and esterases, where bile constituents blend with phospholipids to create bile-phospholipid-mixed micelles ensuing in a range of colloidal species [45,76,78]. These colloidal species subsequently support the solubilisation of poorly water-soluble ingested lipids. Lipid-based formulations are able to exploit this natural process of lipid digestion, consequently affording an ideal microenvironment into which a lipophilic drug with a prominent affinity for lipidic milieus can partition by means of the formation of exogenous lipid carriers [78]. The digestion products of the lipid substrate, including the drug, are subsequently transferred to the enterocytes of the intestinal wall for absorption [76]. Hofmann and Borgström [79] first observed the physiological effects of the chemical absorption of exogenous lipids and declared the bile-phospholipid mixed micelles as being responsible for both the solubilisation of poorly water-soluble drugs, as well as the establishment of a concentration gradient for lipid absorption. Furthermore, as stated, the concurrent consumption of a fatty meal with ACTs such as the ART-LUM combination is well recognised and recommended to invoke this described “food effect” [41,43]. Thus, by including ACTs in a lipid-based formulation, absorption of these drugs may be significantly enhanced from the GIT by increasing the dissolution rate through the facilitation of the formation of colloidal species (solubilised phases). Consequently, drug up-take, efflux and disposition are changed through modification of the enterocyte-based transport [80,81], as well as enhancing drug transport to the systemic circulation and intestinal lymphatic systems [78,82,83]. As described earlier, Kossena et al. [43] have concluded that as little as 2 g lipid is able to stimulate adequate bile salt concentrations to raise intraluminal lipid processing and drug solubilisation. This amount of lipid equates to only two capsules containing a lipid-based formulation to be administered in the fasted state, which is deemed ample to attain the anticipated physiological response [78].

To summarise, the favourable mechanisms of lipophilic drug-uptake, once it is solubilised, include:Facilitated transcellular absorption of the drug as a result of increased membrane fluidity;Paracellular transport of the drug by opening tight junctions;Increased intracellular drug concentration and residence time by surfactants due to the inhibition of P-glycoprotein and/or cytochrome (CYP) P 450 enzymes; andLipid stimulation of lipoprotein/chylomicron production for lymphatic uptake [84].

The lymphatic system is an interesting pathway for drug-uptake. Lipid-based formulations have significant access to this system (Figure 1), which is able to safeguard a lipophilic drug from enzymatic hydrolysis in the GIT wall as well as to circumvent first-pass metabolism, consequently increasing drug bioavailability [85,86]. Lymphatic transport furthermore has the advantage of contributing to site-specific targeting of diseases that manifest in the lymph [56,87]. It plays a distinct role in the transport of drugs to the systemic circulation, given its broad drainage network through the whole body in proximity to the circulatory system [56,87]. The lymphatic system primarily returns products that have escaped into the interstitial space, including oxygen, sugars, proteins and lymphocytes, to the systemic circulation. It also transports essential immune cells around the body. Moreover, it may similarly contribute to the absorption of lipidic products—specifically long-chain fatty acids and lipophilic vitamins—post-digestion [87].

For lipophilic drugs to access the lymphatic pathway, they need to associate with chylomicrons within the enterocytes [78,88]. Charman and Stella [89] found that highly lipophilic drugs (Log P > 5) and long-chain triglycerides with solubilities greater than 50 mg/kg (or 450 mg/mL) can theoretically exploit the lymphatic system as these drugs will travel across enterocytes and associate with enterocyte lipoproteins to form chylomicrons. Following this, the drug-comprising chylomicrons enter the mesenteric lymph duct, move to the thoracic duct and finally enter into the systemic circulation at the junction of the left jugular and left subclavian veins subsequently evading first-pass metabolism [87]. A study conducted by Caliph and co-workers [90] found that the degree of lymphatic transport of the highly lipophilic antimalarial drug, halofantrine (log P 8.5; triglyceride solubility > 50 mg/kg), could notably be associated with triglyceride content in the lymph, signifying that lipid digestion products are indeed required to stimulate the production of chylomicrons.

For these reasons, it can be argued that lipid-based formulations may be utilised to imitate the physiological conditions needed for adequate lymphatic drug delivery. However, these formulations require specific considerations to form acceptable drug delivery systems [91,92]. These requirements, with particular focus on SEDDSs, will be discussed in the following section.

## 4. Self-Emulsifying Drug Delivery Systems (SEDDS) as a Means to Enhance Oral Drug Delivery

The concept of self-emulsifying systems originated in the early 1960s when mixtures of lipid(s) and hydrophilic excipients were employed to solubilise hydrophobic pesticides and chemicals [93]. After two decades, Pouton [93] reported, for the first time, the development of a lipid-based formulation for overcoming various difficulties experienced with lipophilic drugs, namely SEDDS. Thereafter, the prospect of SEDDSs enhancing the bioavailability of lipophilic drugs has been explored by numerous researchers, resulting in the transfer of laboratory research to pharmaceutical industrial set-ups [94].

SEDDSs are unique, isotropic mixtures of drug(s), oil(s), surfactant(s) and co-surfactant(s) that spontaneously form oil-in-water (O/W) emulsions when dispersed in an aqueous phase, normally the GIT environment, under mild agitation [95,96]. These formulation systems were originally developed to overcome the limitations of highly lipophilic drugs (i.e., BCS Class II drugs) such as ART and LUM, and to improve the effective potential delivery of BCS Class III and IV drugs, as well as drugs susceptible to hydrolysis [72,97,98]. The production of SEDDSs is a fairly straightforward process that involves a more effortless formulation technique compared to that of general emulsions. It merely requires the mixing of oil(s), surfactant(s) and/or co-surfactant(s), followed by the addition of a drug to the mixture and subsequently vortexing it until a transparent mixture is obtained. In some instances, the drug may first be dissolved in the excipient/s before adding and mixing the remaining components. Heating is usually also required to obtain a clear, homogenous solution during formulation [65,70,99]. Figure 2 illustrates the uncomplicated manner in which SEDDSs may be manufactured.

SEDDSs allow highly lipophilic drugs to remain in a dissolved state until they are absorbed, thereby overcoming solubilisation challenges and ultimately improving drug delivery [100]. Additionally, SEDDSs permit the entry of drugs into the circulation via the lymphatic pathway [101]—a part of the vascular system that plays a crucial role in the immune system. A few other advantages of SEDDSs include that they:Are comparatively easier to manufacture and scale-up;Depict a proportional increased drug-loading capacity;Generally protect the incorporated drugs from the hostile environment in the gut;Are able to deliver peptides that are prone to enzymatic hydrolysis in the GIT;Allow selective drug targeting towards specific absorption windows in the GIT to be obtained;Do not influence the natural lipid digestion process;Require lower drug doses compared to other traditional formulations given their consistent temporal drug absorption rate profiles; andResult in constant drug absorption, enhanced controlled drug delivery profiles, reduced drug absorption variability and consequently increased oral drug bioavailability [102].

SEDDS, however, is considered an all-inclusive term for certain lipid-based drug delivery systems and should ideally be differentiated into three main categories based on droplet size. The first class is the typical SEDDSs that produce opaque emulsions with a droplet size larger than 300 nm. The other two classes are termed self-micro-emulsifying drug delivery systems (SMEDDSs) and self-nanoemulsifying drug delivery systems (SNEDDSs), which are optically clear emulsions with droplet size ranges of 100–250 nm, and smaller than 100 nm, respectively. These three classes also differ in composition. SEDDSs typically consist of an oil concentration ranging between 40–80% and are produced by including hydrophobic surfactants with hydrophilic-lipophilic balance (HLB) values smaller than 12. SMEDDSs and SNEDDSs, contrarily, comprise an oil phase of less than 20% and are manufactured by including hydrophilic surfactants with HLB values larger than 12. Yet another distinction exists with respect to the mixing process. SNEDDSs will only form when the surfactant and oil phases are mixed first, followed by the addition of water. SMEDDS, on the other hand, will form despite the order in which ingredients are mixed. The input of energy required to form an emulsion is likewise a discerning aspect between SNEDDS and SMEDDS. SNEDDSs necessitate an input of energy, either by mechanical interference or the chemical potential found within the constituents, whereas SMEDDSs originate mainly due to an ultra-low interfacial tension, which is usually achieved by using two or more surfactants/co-surfactants [72,91,103].

SMEDDSs and SNEDDSs, nonetheless, also have some shortcomings in comparison to SEDDS, including higher production costs, lower drug loading and frequently irreversible drug/excipient precipitation that may be difficult to overcome. The high amounts of surfactants included in these formulations are even more concerning as they may prompt GI irritation [91,102,104]. On the positive side, however, smaller oil droplet sizes increase drug bioavailability as a result of an enlarged surface area [105,106]. Moreover, SNEDDS have been found to naturally circumvent first-pass metabolism due to lymphatic absorption [91].

A systematic developmental approach is needed in order to optimise SEDDSs. For this reason, pseudo-ternary phase diagrams are constructed to provide a schematic representation of the region of self-emulsification when specific excipients are employed in combination [107,108]. This basic process entails preparing an augmented surfactant phase (fixed ratio of surfactant to co-surfactant) dissolved within a chosen oil phase in ratios of, for example, 9:1, 8:2, 7:3, 6:4, 5:5, 4:6, 3:7, 2:8 and 1:9. Next, the water phase is included in a dropwise manner whilst slight agitation is initiated through stirring [109,110,111]. The point where the mixture turns cloudy indicates the endpoint as well as the concentration where spontaneous emulsification transpires. Plotting the endpoint on a pseudo-ternary phase diagram demonstrates the concentration range of excipients included in combination as well as where self-emulsification is most likely to occur. Once the coordinates of the endpoint concentrations are plotted within the triangle, the area of spontaneous emulsification is termed heterogeneous due to the biphasic nature of this enclosed region, whereas the unenclosed area indicates the monophasic system known as the homogenous area of the tri-plot [72,107]. Pseudo-ternary phase diagrams are also practical tools employed to predict the phase behaviour of a possible SEDDS, as different sectors of the diagram reveal certain behavioural characteristics of emulsions, as seen in Figure 3. For example, they can illustrate the robustness of a SEDDS to dilution within the GI setting [72,107].

Several attempts have been made to develop an economically viable oral drug delivery system that may overcome absorption restrictions. In this regard, the formulation of SEDDSs (more specifically SMEDDSs and SNEDDSs) have still not reached their full therapeutic potential commercially [112,113], even though a small number of products are already commercially available, as listed in Table 3 [107].

Potentially contributing factors include the limited stability and portability of liquid-SEDDS formulations, propensity for drug crystallisation and precipitation in vivo, low drug loading, poor in vitro-in vivo correlations, as well as high manufacturing costs and distribution processes [103,117]. However, there is hope—the next section will discuss how these and other limitations might still be overcome.

### 4.1. Shortcomings of Oral Liquid Self-Emulsifying Drug Delivery Systems (SEDDSs) and Potential Methods to Overcome Them

Conventional SEDDSs are liquid oral dosage forms, generally used as a liquid measured from a bottle or to fill either soft or hard gelatin capsules. However, as stated, these drug delivery systems display various drawbacks, making the commercialisation of oral liquid SEDDSs challenging [112,113]. For example, Dokania and Joshi [91] indicated that liquid SEDDSs that are incorporated into capsules may filter into and/or interact with the capsule shells. Consequently, the formulator is faced with various notable problems, i.e., the formulation itself may interact with the capsule shells; the shells may become brittle or soft; the capsule shells may leak, leading to insufficient drug loading and subsequent under-dosing; and precipitation of the drug and/or excipients may occur, especially when the dosage form is stored at lower temperatures [118].

During formulation, a surfactant is normally included in a SEDDS to stabilise the drug and aid in the solubilisation thereof [119]. In order to formulate stable SEDDSs, a concentration of approximately 30–60% surfactant is needed as self-emulsification is only obtained once the surfactant concentration surpasses 25%. However, large quantities of these surface-active ingredients may result in GIT irritation. Additionally, if the content of surfactant exceeds 50–60% (depending on the materials used), the emulsification process may be interrupted by the formation of viscous liquid crystalline gels at the oil/water interface, which in turn causes various instabilities [70]. According to the LFCS, Type III lipid-based formulations, such as SEDDSs, include both a hydrophilic surfactant and co-surfactant [61]. The lipid content in these formulations is therefore normally lower than in other lipid-based formulation types, whereas the overall content of the hydrophilic surface-active ingredients is increased. This change in lipophilicity subsequently enhances the risk of drug precipitation and additionally reduces the solubilisation propensity [70].

Other possible instabilities that may present with SEDDSs are irregularities that typically ensue in simple emulsions, which have been described in detail in the preceding research. Briefly, these instabilities include phase separation, phase inversion, creaming, cracking, sedimentation, flocculation, aggregation, coalescence and Ostwald ripening. These are customarily caused by fluctuating temperatures, absence of emulsifiers, dissimilarities in the density of oil and water phases, as well as the high polydispersity index (PDI) values, which indicate the size distribution of droplets in emulsions [65,120,121,122].

Finally, producing these systems is not always considered cost-effective. Proper packaging (utilising the correct capsules and sealing techniques, for example) plays the most significant role in the high production costs of SEDDSs [118,123]. Furthermore, SEDDS formulations theoretically depend on digestion prior to the release of a drug and thus, the dissolution tests normally conducted to conclude pharmaceutical release properties do not suffice [124]. The absence of good predictive in vitro models for the assessment of these formulations is both a practical and economic obstacle for the development of SEDDSs and other lipid-based formulations [84]. Khedekar and Mittal [125] furthermore pointed out that the formulation of SEDDSs containing numerous components (be it drugs and/or excipients) are more challenging to validate.

To prevent the previously mentioned instabilities from arising, a co-surfactant may be combined into the formulation to achieve a resilient fluid interfacial film as well as to reduce the interfacial tension. A highly rigid film is normally formed by the surfactant if no co-surfactant has been added. This, in turn, produces an emulsion with a notably limited range of component concentration. A co-surfactant allows sufficient plasticity of the interfacial film to form different curvatures that are needed to establish an emulsion with a relatively wider composition range [70]. Moreover, several compounds displaying surfactant properties that may be valuable in the formulation of SEDDSs exist. However, the choice is restricted as very few surfactants are tolerated orally [114]. Normally, the incorporation of non-ionic components depicting relatively high HLB values is suggested, as these compounds have been shown to be less toxic compared to ionic surfactants. Nonetheless, they are still able to cause reversible changes to the intestinal lumen permeability [66]. The suggested high HLB value (indicating high hydrophilicity) contributes to the immediate formation of oil-in-water droplets and/or rapid spreading of the formulation in an aqueous media [126], subsequently ensuring a good self-emulsifying/dispersing performance. To attain fast dispersion rates together with small droplet sizes, the optimum HLB values for hydrophilic surfactants and/or co-surfactants were estimated to be between 13 and 15 [127]. This is in line with the literature, which indicated that surface-active agents incorporated into SEDDSs should have an HLB value higher than 12 to be able to provide sufficient phase stabilisation [65]. In addition, by increasing phase stabilisation, drug particles are more effectively covered by the oil-surfactant/co-surfactant phase, thus, maintaining particle integrity and protecting the drug from degradation, which benefits intestinal permeation [128].

### 4.2. Solidification of Self-Emulsifying Drug Delivery Systems (SEDDSs) as an Unconventional Approach in an Attempt to Circumvent Limitations, and the Advantages Thereof

Another means of circumventing some shortcomings is to ensure relatively small droplet sizes and droplet stability. The smaller the droplet size, the smaller the likelihood of cohesion occurring. Droplets of uniform size equally distributed within dispersions are indicative of overall formulation stability [65]. Good droplet stability, and thus, increased emulsion stability, is normally specified by increased negative or positive zeta-potential values (≤−30 mV or >30 mV). These values signify increased electrostatic repulsion between droplets, with higher values being associated with greater emulsion stability. Nevertheless, a small deviation is allowed as emulsions may also be stabilised by both steric and combined electrostatic forces [65]. The careful selection and inclusion of excipients are therefore of immense importance.

A relatively current approach in research to overcome the limitations described is to transform liquid-SEDDS into solid oral dosage forms that impart physicochemical stability and reduce manufacturing and packaging costs while retaining, or augmenting, the pharmacokinetic benefits related to lipids [113,129,130]. Different solidification techniques have been explored to modify liquid SEDDSs into dispersible powders, granules and pellets to improve the stability of these remarkably simplified systems, which are effortlessly and practically up-scaled, rendering several commercial benefits [60,62,65,131]. According to Tang et al. [118], solid SEDDSs combine the advantages of solid oral dosage forms (i.e., high reproducibility and stability) with the advantages of liquid oral SEDDSs (i.e., enhanced bioavailability and solubility [132]), allowing hydrophobic drugs into a fixed-dose combination to reach their full therapeutic potential. In summary, the advantages of solid SEDDS include:Improved patient compliance;High stability and reproducibility;Convenient process control;Thermodynamic stability;Spontaneous formation of liquid SEDDS in the GIT;Faster release rates;Improved solubilisation compared to conventional dosage forms;Selective drug targeting towards a specific absorption window in the GIT;Drug protection from the hostile environment in the gut; andUsage of lipophilic drug compounds that exhibit rate-limiting dissolution steps [130].

Hence, the formulation of solid SEDDS provides a more stable and robust dosage form with lower manufacturing costs comparatively [130]. The next section will deal with solidification techniques that have been investigated in this regard.

## 5. Solidification Techniques

Traditional liquid SEDDS present with a major limitation, namely, poor stability [131]. Researchers have achieved promising results in their efforts to improve this property using various solidification techniques, thereby transforming liquid SEDDS into solid dosage forms [133]. Although various solidification techniques exist, the specific technique chosen should be based on three factors, namely the content of fatty excipients, the properties of the specific drug and the compatibility of the different ingredients in the formulation [131,134]. This section highlights some important solidification techniques that have been described in the literature, including capsule filling, spray drying, spray cooling, melt granulation, melt extrusion and freeze-drying.

### 5.1. Capsule Filling with Liquid and Semisolid SEDDS

As stated previously, the majority of SEDDSs that are currently marketed are composed of liquid formulations filled into either soft or hard gelatin capsules—a process that is considered a means of solidification [113]. Capsule filling is an effortless technique used for the encapsulation of liquid or semisolid self-emulsifying formulations destined for oral use. Highly potent drugs can easily be encapsulated, and relatively high drug loading of up to 50% *w*/*w* can often be achieved, according to Jannin et al. [112]. Moreover, drug loading is only limited by the fill weight and drug solubility [135]. The production of these liquid formulations, which are filled into capsules before being sealed by micro-spraying, requires a much simpler four-step method compared to normal semisolid formulations, i.e.:The semi-solid excipient must be heated to at least 20 °C above its melting point;The active substances are then incorporated into the formulation with a technique called stirring;This resulting molten mixture is subsequently cooled to room temperature;Finally, the cooled “pre-emulsion” mixture is filled into the capsules [91].

This simple manufacturing process is well suited for low dose, highly potent drugs, but one should be mindful of any incompatibilities that can occur between the excipients and the capsule shells [69]. Additionally, storage temperature is a significant consideration in avoiding drug precipitation at decreased temperatures. The most pronounced drawback is the reduction in manufacturing capacity and speed, with manufacturing costs being relatively higher in comparison to normal tablets [136,137]. The best-known soft gelatin capsule examples containing SEDDSs are Neoral^®^, Norvir^®^, and Rocaltrol^®^; whereas illustrations of hard gelatin capsules include Lipirex^®^ and Gengraf^®^ [113].

### 5.2. Adsorption to Solid Carriers

Physical adsorption of liquid SEDDSs on the surface of solid carriers is a simple method to achieve solid lipid formulations, dry emulsions and solid SEDDSs that are free-flowing powders for oral drug delivery [129,138]. Typical adsorbents can be resourcefully used to absorb liquid SEDDS formulations, converting them into a solid form. The solid carriers can be microporous, colloidal inorganic substances, cross-linked polymers or nanoparticle adsorbents. Examples include:Silica, silicates, micronised or amorphous silica (Sylysia^®^) with different grades of pore volume, or silicon dioxide, which is fumed silica (Aerosil^®^) with different grades of specific surface properties;Magnesium trisilicate, magnesium hydroxide or magnesium aluminometasilicate (Neusilin^®^) that may also depict different surface properties (alkaline or neutral) as well as particle size;Calcium silicate; andPorous dibasic calcium phosphate anhydrous (Fujicalin^®^).Other examples are talcum crospovidone, cross-linked sodium carboxymethyl cellulose and/or cross-linked polymethyl methacrylate [129,131].

Cross-linked polymers can moreover provide a favourable environment by enhancing drug dissolution and slowing down drug re-precipitation [91]. Adsorption to solid carriers involves only two to three simple steps, namely: the addition of liquid SEDDS onto the solid carrier by mixing in a blender, followed by filling a capsule with the obtained solid mixture or adding a suitable excipient before compressing it into tablets. Additionally, the porosity of the selected adsorbent material should be carefully deliberated as the length and shape of the pores play a crucial role in the dissolution behaviour of the solid SEDDSs together with the specific surface area [131,139].

This solidification method has the benefits of not utilising organic solvents and only require a few excipients for producing a final formulation. Furthermore, high content uniformity together with significantly high drug loading is obtained, and only basic equipment is employed [112,131,139,140,141]. Nonetheless, prior to formulating, the impact of solidification on the solubilisation capacity and dissolution mechanism of the formulation should be considered [113].

Possible interactions between certain carriers and the included drug, oil/lipid and/or surfactant may furthermore affect the drug dissolution rate of the solid SEDDSs. It is therefore extremely important to sensibly consider the type and amount of adsorbent to be incorporated, as well as the particle size and the specific surface area of the adsorbent, the crystallinity of the drug (the amorphous form is more prone to re-crystallisation and chemical instabilities) and subsequent formulation stability [131,141,142,143].

### 5.3. Wet or Melt Granulation

Another method for solidification of SEDDSs that has been described in the literature is wet or melt granulation, which is a continuous process that provides uniform dispersion of fine particles with no prerequisite on the compressibility of the drug. It accommodates varying pH and moisture levels and is considered safe for human consumption due to the non-swellable and water-insoluble nature of the product. However, this method requires high energy input and cannot be applied to heat-sensitive materials [118,129].

This form of wet/melt granulation follows almost the same steps as normal wet granulation. In this instance, however, the SEDDS formulation acts as a liquid binder normally included in a mixture prior to granulation, where it is melted or softened at relatively low temperatures [117,131,144]. Wet or melt granulation is achieved through a novel technique called fluidised hot melt granulation [145]. The “binder liquid” or SEDDS may be blended with a powder mixture, and the friction of the particles causes heat to be generated during high shear mixing. This forms liquid bridges, which transform into agglomerates or granules [117,131,144]. A carrier such as mannitol, microcrystalline cellulose, lactose, Aerosil^®^ 200, etc., may additionally be utilised to prepare the granulates [101,146,147].

Franceschinis et al. [147], using a high shear mixer, studied the influence of process variables on the properties of self-emulsifying granules. They incorporated a simvastatin-loaded microemulsion as binder, which was dripped onto a powder mixture comprising 70% (*w*/*w*) microcrystalline cellulose, 27% (*w*/*w*) lactose and 3% (*w*/*w*) polyvinyl pyrrolidone. As this “granulating liquid” did not only include water, but also contained an oily phase, more plastic/deformable granules depicting smaller sizes were created comparatively to the granules produced via a mere water-based granulate. These granules displayed longer disintegration times due to the presence of oily bridges that decreased their wettability as the consolidation of the granules was more extensive. The researchers concluded that the choice of the most adequate SEDDS formulation to be processed into self-emulsifying granules should not only be based on drug solubility and droplet size of the specific emulsion formed upon reconstitution, but the viscosity of the preliminary emulsions/microemulsions should also be contemplated as it should ideally be low enough to enable dripping of the granulating fluid.

Tang et al. [118] suggested that this type of granulation process is controlled by a few main parameters, including impeller speed, mixing time, binding particular size, the thermoplastic behaviour of the binder, as well as the viscosity of the “binder” during granulation. Additionally, it has been found that the viscosity of the SEDDSs or “meltable binders” largely affects nucleation and particle size at high and low impeller speeds, respectively. “Binders” with lower melting points risk the continuous melting and softening of the agglomerates during handling and storage, whereas binders with higher melting points necessitate high temperatures, which can lead to instability, especially for heat-sensitive materials. Two additional mechanisms, namely distribution and immersion, play a crucial role in the development of granules by means of wet or melt granulation. Generally, SEDDSs that contain lipids with low HLB values and high melting points are most suitable for sustained release applications, whereas excipients portraying high HLB values can be utilised in immediate release dosage forms where bioavailability is enhanced [112,118,125,129].

### 5.4. Melt Extrusion or Extrusion-Spheronisation

Melt extrusion or extrusion-spheronisation is considered one of the most investigated methods for self-emulsifying pellet production. Extrusion is a solvent-free process that converts raw material with plastic properties into an agglomerate that displays cylindrically shaped granules with a uniform density. This is achieved through mixing of the SEDDS with a carrier or adsorbent excipient, followed by agglomeration and successive spheronisation, where the extrudate is broken down and formed into spherical particles. These spheroids or pellets usually depict a narrow particle size distribution, good flowability and low friability. The process is furthermore said to allow for high drug loading [125,131,138].

Self-emulsifying pellet formation is mostly influenced by the composition of the pellets. An ideal pellet will consist of the smallest possible amount of adsorbent material and the largest possible amount of liquid SEDDS, thus obtaining pellets with acceptable physical characteristics and maximum drug loading. Therefore, the amount of loaded SEDDS depends significantly on the ratio of the components. A larger amount of the self-emulsifying mixture can be included in the formulation of a larger amount of the absorbent material is incorporated relative to the amounts of other excipients [148,149,150]. However, Abdalla et al. [151] reported that increasing the load of SEDDS in pellets to higher than 40% may cause sticking of the extrudate to the apparatus. In the absence of adsorbents such as microcrystalline cellulose, pellets may display poor flowability, a higher propensity to agglomerate and low pellet hardness. It was furthermore found that pellet tensile strength and irregularity directly depend on the relative amounts of absorbent and other excipients included, as well as the ratio of SEDDS:water. The amount of included SEDDS positively relates to the required extrusion force and the roundedness of the pellets. However, it also leads to higher friability of the said pellets, probably due to abridged interactions within the pellets themselves [148,149,150,151,152]. Nonetheless, an appropriate binding agent, such as polyvinyl pyrrolidone, can supposedly improve the mechanical strength of the formed self-emulsifying pellets [153]. Moreover, the inclusion of a disintegrant provides the added benefit of rapid drug release [148,149,150]. Abdalla and Mäder [148] furthermore found that the order in which the various components were mixed significantly influenced the successful melt extrusion process.

Melt extrusion operates with small equipment; it offers high mixing efficiency and prevents cross-contamination as it is a closed process. The processing time is short and has easily controlled parameters. In contrast, melt extrusion has some drawbacks: a high-energy input is needed, heat-sensitive materials cannot be utilised, only a limited number of polymers may be employed and various properties of the components should first be intensively researched (for example, stability, flow properties, wettability, chemical and physical interactions, etc.). Despite these drawbacks, it has been concluded that pellets produced by melt extrusion are capable of maintaining the properties of liquid SEDDS, rendering this method appropriate for the production of solid multi-unit SEDDSs [150,153,154].

### 5.5. Freeze-Drying

Freeze-drying, also known as lyophilisation or cryodesiccation, is still an extremely novel solidification technique for liquid or semi-solid SEDDS ingredients. For this reason, literature available on this subject is still limited. In short, freeze-drying is a process that allows sublimation of a frozen aqueous phase present in liquid self-emulsifying formulations at reduced temperatures and pressures to obtain a powder that, upon reconstitution with an aqueous phase, produces a fine micro- or nano-emulsion [94,155]. This is thus a drying technique where the product is solidified by freezing before evaporating the solvent via sublimation to obtain a lyophilised molecular dispersion. Cryoprotectants are occasionally added to prevent the minimal stresses that occur in the lipid bilayers during the freezing and drying processes. Numerous sugars, such as trehalose, sucrose, maltose, glucose and mannitol, have been used as cryoprotectants for emulsified systems [155]. The lyophilised SEDDS offers the advantages of improved drug stability, efficacy, handling and patient compliance. This technique has been explored as an alternative approach to the production of dry emulsions from oil-in-water emulsions [156] and its application in the manufacturing of solid dispersions has delivered promising results. A typical freeze-drying cycle consists of only three stages: freezing, primary drying and secondary drying. In theory, it is thus a relatively simple and fast technique to solidify SEDDSs [134,155].

### 5.6. Spray-Drying

Spray drying is considered an unpretentious and economical single-step method for fabricating micro-particulate drug delivery systems and can easily be scaled up. Likewise, it has the auspicious ability to solidify SEDDSs that may afterwards be manufactured into oral tablets. This technique involves solubilising mixtures composed of drugs, lipids, surfactants (i.e., SEDDSs) and solid carriers before subjecting the formulation to atomisation, which results in a spray of droplets. Throughout this process, the volatile phase (water in the emulsion) evaporates as the droplets are introduced into a drying chamber, forming dry particles in a controlled environment [94,113,131,157,158,159,160]. Singh et al. [161] compared freeze-dried and spray-dried solid-SEDDSs containing valsartan. The spray-dried solid-SEDDS depicted a significantly more uniform particle size distribution, notably improved powder flowability as well as fast GI processing, with a slight enhancement in the overall pharmacokinetic performance (approximately 5% higher oral bioavailability).

As previously mentioned, the production process of solid SEDDS is made possible through the inclusion of a carrier—be it hydrophilic or lipophilic. The choice of this carrier notably affects drug release as well as oral drug bioavailability as it influences SEDDS entrapment efficiency and droplet size of the emulsion formed during reconstitution. Hydrophilic carriers, such as hydroxypropyl cellulose, hydroxypropyl methylcellulose, lactose, maltodextrin and microcrystalline cellulose, allow for faster water penetration, which in turn permits prompt microemulsion formation. Crospovidone and silica are contrariwise examples of solid hydrophobic polymers utilised as carriers. However, the type of carrier is not the key factor affecting SEDDS loading, flowability, droplet size upon re-formation, drug precipitation and/or self-emulsifying properties [94,131,157,159]. Other factors that contribute considerably are the:Types and combinations of excipients incorporated in both the SEDDS formulation as well as during the spray-drying process;Porous structure and distribution of the carrier included and subsequent entrapment efficiency [159,162];Viscosity of the gel layer that may form around the solid particles during reconstitution due to the incorporated carrier [163];Emulsifier concentration that may be responsible for poorer self-dispersability [133,160]; andBatch-to-batch variability relating to the natural origin of included excipients, which may significantly influence the quality of a product [131].

By introducing, for example, gelatin into the formulation during spray-drying, drug precipitation during reconstitution may be circumvented through the gelatin’s ability to block active surfaces and provide steric stabilisation, thereby averting drug nucleation and crystal growth in the supersaturated state of the drug in the medium [164,165]. Certain process parameters additionally impact the self-emulsifying properties of solid SEDDSs [159]. It is therefore critical to maintain the inlet and outlet air temperatures, the feed temperature, the volatility of the solvent and the nozzle material [118].

Spray-drying has many advantages. It is not only time-saving and adaptable to fully automated control systems, but can also be implemented in both heat-resistant and -sensitive products. Furthermore, it provides high precision control in particle size, bulk density, degree of crystallinity, organic volatile impurities and residual solvents. Spray-drying also offers constant powder quality throughout the process of drying, reduces corrosion and is applicable in various industries, ranging from aseptic pharmaceutical processing to ceramic powder production [40]. On the downside, a lower yield may be obtained due to the removal of non-encapsulated free drug with the exhausted gas [160,162], and process yield (effectiveness of SEDDS entrapment) may also be negatively affected by the type of carrier used [131].

### 5.7. Spray-Congealing or Spray-Cooling

Spray-congealing involves spraying a molten oily formulation through a cooling chamber, upon which liquefied droplets solidify and re-crystallise into spherical fine powders. These congealed particles are strong and non-porous due to the absence of solvent evaporation. The fine powder may subsequently be processed into various solid oral dosage forms by directly filling the powder into hard gelatin capsules or by compressing it into tablets [166]. This method does, however, necessitate the installation of larger, more expensive industrial apparatus compared to other tooling employed to obtain pharmaceutical powders for tableting. Even so, this method is continuous, easy and adaptable to automatic operations with fast response times. The powder quality remains constant throughout the cooling process regardless of the time when cooling conditions remain constant. While rotary, pressure or two-fluid atomisers can be used to pulverise the liquid mixture and produce particles, ultrasonic atomisers are usually preferred. In addition, defined and narrow melting points are preferable when considering the selection of meltable materials to be incorporated for spray-congealing [167].

### 5.8. Supercritical Fluid-Based Methods

This method employs lipids for preparing solid dispersions or for coating drug particles. The coating procedure involves dispersion of the drug particles in a supercritical fluid containing one or more coating material(s) [168]. Generally, the preferred supercritical fluid of choice is carbon dioxide. The benefits include the fact that it is non-toxic, non-flammable, easy to recycle, cost-effective and possesses environmentally acceptable characteristics [169]. The solubility of the coating material is supported by elevated temperature and pressure. Subsequently, the coating process is further assisted by the gradual decrease in the solubility of the coating material leading to its gradual displacement onto drug particles. Several factors that are considered during this formulation method are the:Solubility of the active substance and excipients in the supercritical fluid;The energy and environmental settings regarding the evaporation of solvents; andThe stability of the active substance under the formulation conditions [168].

Supercritical fluid-based methods are advisable for highly potent, low-dose drugs due to the favourable potential for lipid exposure and a comparatively lower drug loading capacity [170].

## 6. Solid Self-Emulsifying Drug Delivery Systems May Pose a Bright Future for the Fight against Malaria

Malaria continues to be a widespread infectious disease, placing immense pressure on world health [171,172,173,174,175,176]. As emphasised through numerous studies, it is one of the most serious vector-borne diseases prevalent in 104 tropical and subtropical regions. It furthermore contributes significantly to the poverty cycle, as the highest global disease burden is concentrated mainly in Africa, accounting for approximately 90% of the reported international cases [2,177,178,179]. Of these, mainly young children in stable transmission areas endure the highest malarial morbidity and mortality affliction [180]. In 2019, the WHO reported that the malaria mortality rate in children equated to a child dying every two minutes and that each year, more than 200 million new cases of the disease are recounted [2,4]. It disproportionally affects the lowest income and most vulnerable populations that have restricted access to sufficient health care services and has an injurious effect on education through missed school days due to illness [27]. The WHO additionally highlighted that the world is at a critical point in the fight against malaria and an opportunity and urgent need exist to decrease morbidity and mortality by identifying approaches that aim to reduce transmission [2,17,27]. Daniel Vasella, the chief executive of Novartis, stated: “The fight against malaria is a complex one. Availability of the drug is only one element” [176]. Moreover, resistance towards malaria treatment is now also considered a global threat [181]. However, as stated, it seems that incipient drug resistance is not the main concern, but rather under-dosing occurrences [54]. This is due to the inability to effectively deliver highly lipophilic treatment regimens (i.e., varied and/or limited drug absorption), thus needing a fatty meal in order to achieve adequate therapeutic drug levels in patients too nauseous to eat, for a period long enough to attain full recovery [44]. It is furthermore very difficult for patients residing in a third world country, that are not only constantly nauseous, but who live in poverty-stricken areas and do not have the means to obtain proper nutrition [182]. It can consequently be argued that under-dosing rather stems most probably from an inadequate drug delivery system.

Although new chemical entities are being developed, progress is slow due to a lack of funding, and the time it takes to develop novel dosage forms is torturous. For these reasons, scientists are researching methods to improve existing drugs as well as their drug delivery systems [173]. In an effort to conquer the fight against malaria, the WHO has called for the development and implementation of oral ACTs to combat resistance as they have shown to significantly reduce the mortality rate of malaria over the past few years [4,183,184,185,186,187,188]. Pharmaceutical formulation scientists regularly face the problem of insufficient drug solubility as it directly affects the therapeutic ability of a drug. Currently, most novel drugs synthesised are similar to artemether and lumefantrine, i.e., highly lipophilic in nature and displaying poor aqueous solubility, highlighting the predicament we are in [189]. Thus, poor bioavailability, inadequate therapeutic drug-blood levels, poor water solubility, patient noncompliance and often the short half-lives of orally administered drugs have driven researchers to rather advance drug delivery systems that are able to improve the therapeutic efficacy of these types of drugs [39,40,44,190].

Fortunately, among the numerous technologies that are accessible to improve solubility and overcome bioavailability issues, economically viable lipid formulations have been extensively employed in accepted oral drug products [56,132]. SEDDSs, for example, have become a promising strategy that allows improvement in the formulation of oral drug delivery systems employing active ingredients with low aqueous solubility characteristics (BCS class II); in addition, it is able to provide the benefit of improved bioavailability. However, to date, no anti-malarial drug combination has been formulated into specialised oral lipophilic drug delivery systems such as SEDDSs. Moreover, it has been proven that lipid quantities as low as 2 g are necessary to adequately solubilise lipophilic drugs, subsequently rendering SEDDSs able to facilitate in the matter that fatty meals are essential in aiding antimalarial drug solubilisation and absorption [41,48,49]. Despite their popularity, however, the stability of liquid SEDDS present some obstacles, including a tendency for drug crystallisation and precipitation, low drug loading, as well as poor in vitro-in vivo correlations [91,191]. In conclusion, various solidification techniques of SEDDS have recently been subject to investigation and have indeed proven superior to other techniques by simplifying manufacturing and reducing production costs (as these techniques can easily be up-scaled due to already existing industrial equipment that could be utilised), as well as improving product stability and patient compliance [113,128]. Solid SEDDSs appear to have a bright future, especially in the fight against malaria, and will hopefully continue to enable the incorporation of highly hydrophobic drugs into lipophilic drug delivery systems.

## Figures and Tables

**Figure 1 pharmaceuticals-15-00120-f001:**
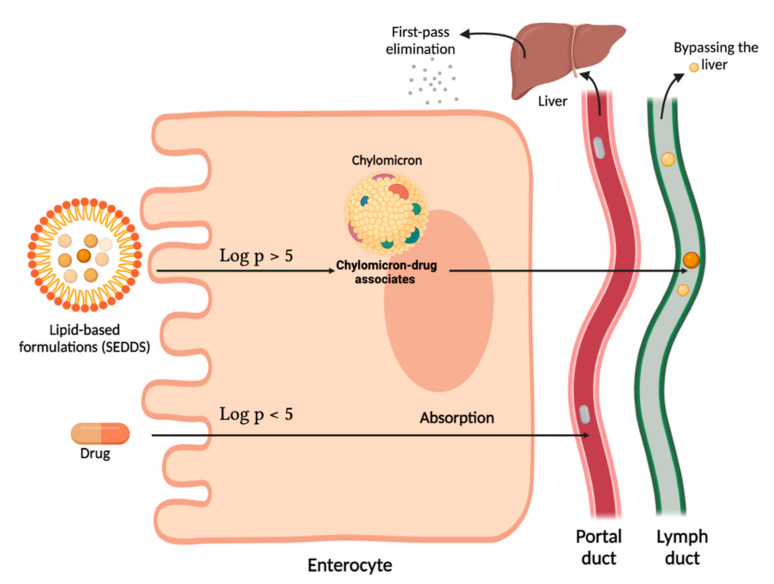
Mechanism of lipid-based formulations by means of the lymphatic system circumventing first-pass metabolism.

**Figure 2 pharmaceuticals-15-00120-f002:**
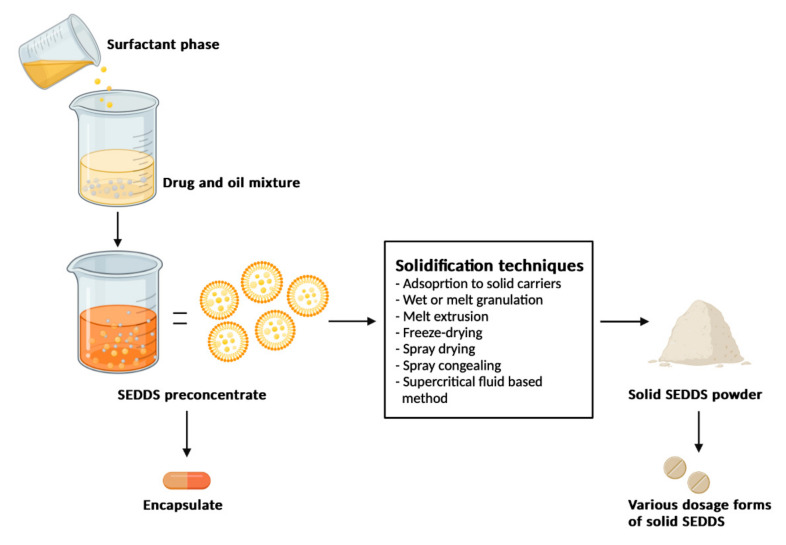
Manufacturing of self-emulsifying drug delivery systems (SEDDS) is an uncomplicated and relatively fast process, which may be followed by various solidification techniques and subsequent processing.

**Figure 3 pharmaceuticals-15-00120-f003:**
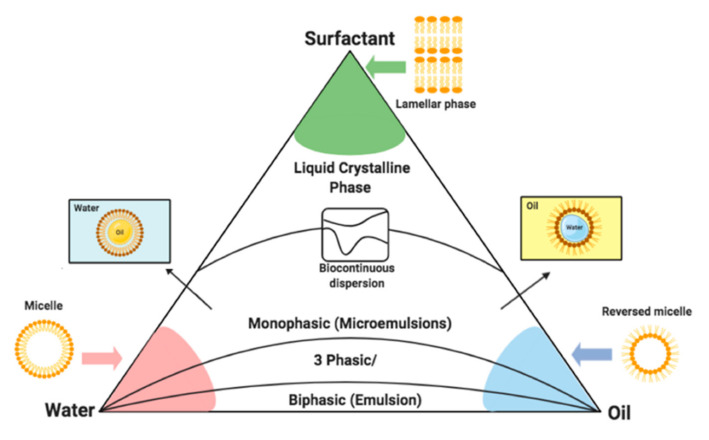
Hypothetical pseudo-ternary phase diagram demonstrating phase behavior exhibited by emulsions.

**Table 1 pharmaceuticals-15-00120-t001:** Treatment regimens for uncomplicated malaria (adapted from [12]).

*Plasmodium* Parasite	First-Line Treatment	Substitute Treatment
*P. vivax* *P. ovale* *P. cynomolgi*	Chloroquine At first, 10 mg/kg; then after 12, 24, 36 h 5 mg/kg OR Dihydroartemisinin/piperaquine followed by (post G6PD-deficiency exclusion) Primaquine 0.5 mg/kg/day (max. 30 mg); 14 days OR Tafenoquine (patients ≥ 16 years) 300 mg single dose	Artemether/lumefantrine OR Atovaquone/proguanil OR Mefloquine followed by (post G6PD-deficiency exclusion) Primaquine 0.5 mg/kg/day (max. 30 mg); 14 days + Chloroquine Adults: 100–150 mg/day; 14 days Children: 25–75 mg/day; 14 days OR Tafenoquine (patients ≥ 16 years) 300 mg single dose
*P. malariae*	Chloroquine Initially 10 mg/kg; followed by 5 mg/kg after 12, 24, 36 h	Artemether/lumefantrine OR Atovaquone/proguanil OR Dihydroartemisinin/piperaquine OR Mefloquine
*P. falciparum*	Artemether/lumefantrine 2 doses/day for 3–5 days OR Dihydroartemisinin/piperaquine 1 dose/day for 3 days	Atovaquone/proguanil 1 dose/day for 3 days OR Mefloquine 25 mg/kg in 1–4 doses (if ≥ 2 doses separate doses by 6–8 h)
*P. knowlesi*	Artemether/lumefantrine OR Dihydroartemisinin/piperaquine	Chloroquine If unavailable: Atovaquone/proguanil OR Mefloquine

**Table 2 pharmaceuticals-15-00120-t002:** Recommended artemisinin-based combination therapy (ACT) regimens for uncomplicated *Plasmodium falciparum* malaria.

ACT	Commercial Products
**Artemether-lumefantrine** 20:120 mg—dispersible tablets 40:240 mg—standard tablets	Coartem^®^ Riamet^®^ Falcynate-LF^®^ Faverid^®^ Amatem^®^ Lonart^®^
**Artesunate-amodiaquine** 25:67.5 mg—standard tablets 50:135 mg—standard tablets 100:270 mg—standard tablets	Coarsucam^®^
**Dihydroartemisinin-piperaquine** 20:160—paediatric tablets 40:320 mg—standard tablets	Duo-Cotecxin^®^ Artekin^®^ Eurartesim^®^
**Artesunate-mefloquine** 25:50 mg base—paediatric tablets 100:200 mg base—standard tablets	Artequin^®^ ASMQ^®^
**Artesunate with sulfadoxine-pyrimethamine** 50 mg scored artesunate tablets & 500:25 mg sulfadoxine/pyrimethamine fixed-dose combination tablets	Amalar plus^®^

**Table 3 pharmaceuticals-15-00120-t003:** Some commercially available self-emulsifying drug delivery systems (SEDDSs) [107,114,115,116].

Product Name and API ^1^	Dosage Form	Application	Manufacturer
Accutane^®^ (Isotretinoin)	SGC ^2^	Severe refractory nodular acne	Hoffmann-La Roche Inc. (Basel, Switzerland)
Agenerase^®^ (Amprenavir)	SGC	Protease inhibitor used to treat HIV-1 infections	Glaxo Group Ltd. (Brentford, UK)
Aptivus^®^ (Tipranavir)	SGC	Combination antiretroviral treatment of HIV-1 strains resistant to more than one protease inhibitor in patients who are treatment-experienced	BoehringerIngelheim Pharmaceuticals Inc. (Ingelheim, Germany)
Convulex^®^ (Valproic acid)	SGC	Antiepileptic	Gerot Lannach (Lannach, Austria)
Depakene^®^ (Valproic acid)	SGC	Used as mono- and adjunctive therapy in the treatment of patients, 10 years and older, with simple and complex partial seizures that occur either in isolation or in association with other types of seizures	AbbVie Inc. (Lake Bluff, FL, USA)
Fortovase^®^ (Saquinavir)	SGC	Protease inhibitor used to treat HIV-1 infections	Hoffmann-La Roche Inc. (Basel, Switzerland)
Gengraf^®^ (Cyclosporine)	HGC ^3^	A systemic immunosuppressant to treat rheumatoid arthritis and psoriasis	AbbVie Inc. (Lake Bluff, FL, USA)
Lipirex^®^ (Fenofibrate)	HGC	Treatment of: Severe hypertriglyceridemia with or without low HDL cholesterol, Mixed hyperlipidemia when statins are contraindicated/avoided, Mixed hyperlipidaemia with high cardiovascular risk, when triglycerides and HDL cholesterol are not amply controlled	Highnoon Laboratories Ltd. (Lahore, Pakistan)
Neora^®^ (Cyclosporine)	SGC	Systemic immunosuppressant	Novartis (Basel, Switzerland)
Norvir^®^ (Ritonavir)	SGC	Protease inhibitor used to treat HIV-1 infections	Highnoon Laboratories Ltd. (Lahore, Pakistan)
Rocaltrol^®^ (Calcitriol)	SGC	Treatment of: Secondary hyperparathyroidism, Hypocalcaemia and abnormal phosphate metabolism in patients with renal osteodystrophy, Established post-menopausal osteoporosis	Validus Pharmaceuticals LLC. (Parsippany-Troy Hills, NJ, USA)
Sandimmune Neoral^®^ (Cyclosporine)	SGC	Treatment of: Rheumatoid arthritis, Psoriasis, Organ rejection prophylaxis, Certain Orphan Designations	Novartis (Basel, Switzerland)
Targretin^®^ (Bexarotene)	SGC	Treatment of cutaneous T-cell lymphoma in patients depicting resistance to at least 1 systemic drug regime	Ligand Pharmaceuticals/Eisai Ltd. (Tokyo, Japan)
Vesanoid^®^ (Tretinoin)	SGC	Management of APL-acute promyelocytic leukaemia	Hoffmann-La Roche (Basel, Switzerland)

^1^ API—active pharmaceutical ingredient; ^2^ SGC—soft gelatin capsules; ^3^ HGC—hard gelatin capsules.

## Data Availability

Data is contained within the article.

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
