# Peer review of "Solidification of Self-Emulsifying Drug Delivery Systems as a Novel Approach to the Management of Uncomplicated Malaria"

_pharmaceuticals, 2022, doi:10.3390/ph15020120_

Round 1
Reviewer 1 Report
The manuscript under consideration presents very interesting review on the drug delivery systems for malaria treatment. The review is well-written and would be interesting for the wide range of scientists.
I would only recommend before publishing enrich the review with some extra illustrations e.g. scheme of synthesis of Self-Emulsifying Drug Delivery Systems. This will significantly increase the level of manuscript
Author Response
Dear Reviewer,
Thank you very much for this suggestion, we included a schematic representation of synthesis of Self-Emulsifying Drug Delivery Systems. This is indeed a great suggestion and will most definitely add value to the work done. Much appreciated!
Please see Page 10, Lines 383–388, Figure 2 included in the manuscript as track changes in MS Word (red and bold).
- Changed manuscript text (Lines 383–388):
“Figure 2 illustrates the uncomplicated manner in which SEDDSs may be manufactured.”
“Figure 2. Manufacturing of self-emulsifying drug delivery systems (SEDDS) is an uncomplicated and relatively fast process, which may be followed by various solidification techniques and subsequent processing.”
Reviewer 2 Report
The manuscript presents the overview of the novel approaches for oral lipophilic drug delivery systems to the management of malaria, including self-emulsifying drug delivery systems. This is an interesting article that is relevant to pharmaceutical science. However, there are some remarks:
- Page 7, line 285 – authors should clarify abbreviation “GI”
- Page 11, line 455 – the authors are suggested to bring examples of commercially available products of SEDDSs.
- Page 13 – the authors are suggested to highlight chapter 4.2. about advantages of solid SEDDS
- The authors are suggested to use the following reference in their review paper:
- Potharaju, S., Mutyam, Sh.K., Liu, M., Green, C., Frueh, L., Nilsen, A., Pou, S., Winter, R., Riscoe, M.K.m, Shankar, G. Improving solubility and oral bioavailability of a novel antimalarial prodrug: comparing spray-dried dispersions with self-emulsifying drug delivery systems Dev. Tech. 2020, 25 (5), 625-639. doi.org/10.1080/10837450.2020.1725893

Author Response
Thank you kindly for the first comment.
However, there are some remarks:
Point 1:
Page 7, line 285 – authors should clarify abbreviation “GI”.
- Response to reviewer:
Please see page 6, lines 191–192, where the abbreviation was first provided. For your convenience, we have highlighted it by using MS Word track changes (highlighting in bold and red).
- Indicated in manuscript text (Lines 191–192):
“Nonetheless, due to the typical gastrointestinal (GI) disturbances normally experienced with malaria…”
Point 2:
Page 11, line 455 – the authors are suggested to bring examples of commercially available products of SEDDSs.
- Response to reviewer:
Thank you sincerely for this valid suggestion. We have now included a table to indicate various commercially available SEDDS products. Please see Page 12–13, Lines 459–463 as well as a new table (Table 3) now included into the paper.
- Changed manuscript text (Lines 459–463):
“…even though a small number of products are already commercially available as listed in Table 3 [107].”
“Table 3. Some commercially available self-emulsifying drug delivery systems (SEDDSs) [107,114–116].”
Point 3:
Page 13 – the authors are suggested to highlight chapter 4.2. about advantages of solid SEDDS.
- Response to reviewer:
We once again want to thank the Reviewer for this valuable recommendation. The authors have added Section 4.2 to emphasize the advantages of solid SEDDS. Please see Page 15, Lines 535–537, where the change is indicated in red and bold in MS Word, track changes.
- Changed manuscript text (Lines 535–537):
“4.2. Solidificatioin of self-emulsifying drug delivery systems (SEDDSs) as an unconventional approach in an attempt to circumvent limitations; and the advantages thereof”
Point 4:
The authors are suggested to use the following reference in their review paper:
Potharaju, S., Mutyam, Sh.K., Liu, M., Green, C., Frueh, L., Nilsen, A., Pou, S., Winter, R., Riscoe, M.K.m, Shankar, G. Improving solubility and oral bioavailability of a novel antimalarial prodrug: comparing spray-dried dispersions with self-emulsifying drug delivery systems, Dev. Tech. 2020, 25 (5), 625-639. doi.org/10.1080/10837450.2020.1725893.
- Response to reviewer:
The authors agree that this reference will add value to the paper. Again, thank you kindly for the considerate suggestion. Please find the addition of the reference on Page 15, Line 557; Page 21, Line 876, as well as in the Reference List at the end of the paper, specified in MS Word, track changes in red and bold.
- Changed manuscript text (Lines 557 and 876):
“(i.e., enhanced bioavailability and solubility [133])”
“...been extensively employed in accepted oral drug products [56,133].”
Reviewer 3 Report
The manuscript entitled "Solidification of Self-Emulsifying Drug Delivery Systems as a Novel Approach to the Management of Uncomplicated Malaria" reviews the use of SEDDS as carriers for oral delivery of antimalarial agents.
After illustrating advantages and drawbacks of therapeutic agents used in the treatment of malaria, the authors describe the properties of SEDDS and the techniques used to prepare such delivery systems.
The manuscript is well organized although some concepts are repeated different times.
Author Response
Dear Reviewer, thank you very much for the comment.
If there were concepts repeated, it was only to emphasize the importance and to bring all the concepts together in order to form a unified idea.
Reviewer 4 Report
Eun Bin Seo and co-authors have done very good job and the manuscript covers oral delivery of lipophilic drugs specially by means of self-emulsifying drug delivery systems. Authors have covered detailed information about technical aspects concerning designing of SEDDS. I genuinely recommend to accept this manuscript for the publication.
good luck!
Author Response
Dear Reviewer,
Thank you very much for the remark!!
We sincerely appreciate it!